# Development and Investigation of Lanthanum Sulfadiazine with Calcium Stearate and Epoxidised Soyabean Oil as Complex Thermal Stabilizers for Stabilizing Poly(vinyl chloride)

**DOI:** 10.3390/polym11030531

**Published:** 2019-03-21

**Authors:** Qiufeng Ye, Xiaotao Ma, Bobin Li, Zhe Jin, Yingying Xu, Cheng Fang, Xiaoya Zhou, Yeqian Ge, Feng Ye

**Affiliations:** 1College of Textile and Garment, Shaoxing University, Shaoxing 312000, China; yeqiufeng@nimte.ac.cn (Q.Y.); mxt775382687@163.com (X.M.); 18868914618@163.com (Y.X.); 18857557347@163.com (C.F.); m15988319825@163.com (X.Z.); geyeqian@163.com (Y.G.); 2Ningbo Institute of Materials Technology and Engineering, Chinese Academy of Sciences, Ningbo 315000, China; 3Shaoxing Testing Institute of Quality and Technical Supervision, Shaoxing 312366, China; libbsx@126.com (B.L.); jinzhe981@163.com (Z.J.)

**Keywords:** lanthanum sulfadiazine, poly(vinyl chloride), calcium stearate, epoxidised soya bean oil, synergistic effect, thermal decomposition kinetics

## Abstract

Lanthanum sulfadiazine (LaSD) was synthesized from sulfadiazine and lanthanum nitrate using water as solvent under alkaline conditions, and was used as a novel rare earth thermal stabilizer to stabilize poly(vinyl chloride) (PVC). The structure of LaSD was characterized by elemental analysis (EA), Fourier transform infrared spectroscopy (FTIR) and thermo- gravimetric analysis (TGA). The influence of lanthanum sulfadiazine with calcium stearate (CaSt_2_) and epoxidized soybean oil (ESBO) on stabilizing PVC was studied by using the Congo red test, oven discoloration test, UV-vis spectroscopy and thermal decomposition kinetics. The results showed that the addition of LaSD as a thermal stabilizer can significantly improve the initial whiteness and long-term stability of PVC. In addition, the synergies between LaSD, ESBO, and CaSt_2_ can provide outstanding improvement in the long-term thermal stability of PVC. When the ratio of LaSD/ESBO/CaSt_2_ is 1.8/0.6/0.6, its thermal stability time is 2193 s which is the best state for stabilizing PVC. Furthermore, comparing the reaction energy (E_a_) and the variations in the conjugate double bond concentration in PVC samples, the order of thermal stability of PVC was PVC/LaSD/ESBO/CaSt_2_ > PVC/LaSD/ESBO > PVC/LaSD. The thermal stability mechanism of LaSD on PVC was studied by the AgCl precipitation method and FTIR spectrum. The results showed that the action of LaSD on PVC was achieved through replacing unstable chlorine atoms and absorbing hydrogen chloride.

## 1. Introduction

Poly(vinyl chloride) (PVC) is the third largest plastic after polyethylene (PE) and polypropylene (PP) [1,2]. It can be widely used in industry, architecture, agriculture, medicine and other fields because of its outstanding physical and chemical properties, including fire resistance, wear resistance, best comprehensive mechanical properties, transparency, electrical insulation, chemical resistance, heat resistance, sound insulation and other advantages [3,4]. However, PVC will degrade due to its instability at the processing temperature, so thermal stabilizers need to be added during processing [5,6]. Thermal stabilizers can be generally divided into lead salts, soap salts, organotin, rare earth and organic auxiliary thermal stabilizers. The first three have been industrialized and well applied in PVC production [7]. However, although lead salts and organotin stabilizers have a strong stabilizing effect on PVC, their application is limited due to their toxicity and cost limitations [8,9]. Now, the European Union has completely banned the use of lead salt stabilizer in PVC products and organotin stabilizer is only used for machine parts. Metal soap is the most commonly used thermal stabilizer for PVC due to its advantages of low price, non-toxic and high efficiency. The carboxyl group on the metal soap replaces the tertiary chlorine or allyl chlorine atoms in the main chain of PVC, thus effectively inhibiting the dehydrogenation of PVC and improving its initial whiteness. Unfortunately, Ca/Zn stabilizer has some drawbacks with regard to long-term stability due to its significant “zinc-burning” effect [10,11]. Therefore, it is very important for today’s PVC industry to find an efficient thermal stabilizer with both initial stability and long-term stability.

Rare earth complex stabilizers have attracted the attention of many scholars in recent years due to their non-toxic, highly efficient and multi-functional properties [12,13]. The excellent thermal stability of rare earth stabilizers may be due to their ability to inhibit dehydrochlorination and replace the unstable functional groups on the PVC main chain [13]. Among the numerous rare earth thermal stabilizers, organic lanthanum complex stabilizer has a high thermal stability for PVC [14,15]. Therefore, in recent years many researchers have spent a lot of time in preparing new lanthanum complex stabilizers and investigated their effect on the thermal stability of PVC [16,17,18]. Li et al. reported a lanthanum pentaerythritol alkoxide (La-PE), which was synthesized through the traditional preparation methods and they also investigated the synergistic effect between La-PE and ZnSt_2_. The results showed that the La-PE/ZnSt_2_ complex stabilizer showed a good initial color and long-term stability on PVC [16]. Li et al. designed and synthesized a ternary complex (La(His)_2_/PE/ZnSt_2_) and applied it as a stabilizer for PVC. The results revealed that the PVC co-stabilized with La(His)_2_/PE/ZnSt_2_ and had good initial color and long-term stability [17]. Li et al. prepared a basic complex (LaL) by the precipitation method in aqueous solution and investigated its synergistic effect with pentaerythritol (PE). The results also indicated that the PVC co-stabilized with LaL/PE had good initial color and long-term stability [18]. Therefore, the investigation of the interaction between rare earth stabilizers and other auxiliary stabilizers is very important for the development of rare earth stabilizers in the PVC industry.

In recent years, a large number of studies have shown that the presence of uracil in compounds has a positive effect on improving the thermal stability of PVC [19,20,21,22]. Sabaa et al. investigated uracil derivatives as PVC thermal stabilizers through a series of experiments and found that they can significantly improve the thermal stability of PVC [19,20]. Wang et al. designed and synthesized a kind of biuracil derivative with high thermal stability and transparency for PVC [21,22]. The key factor to explain the mechanism of uracil derivatives on the thermal stability of PVC is the amino group, which can replace the unstable chlorine atoms on PVC chain and absorb the HCl released by PVC degradation. On the other hand, sulfadiazine is a heterocyclic compound containing the elements of carbon, nitrogen, oxygen, sulfur and hydrogen. Its molecular formula is C_10_H_10_N_4_O_2_S, which is a derivative of uracil [23,24]. In this paper, lanthanum sulfadiazine synthesized from sulfadiazine was selected as the thermal stabilizer for PVC for the following three reasons. First, sulfadiazine can be converted into lanthanum sulfadiazine as a new thermal stabilizer of lanthanum salt. Second, sulfadiazine has an amino group, which has been proved to play a major role as a thermal stabilizer in PVC. Third, sulfadiazine is an antimicrobial agent with a broad spectrum of antimicrobial activity. Based on these three features of sulfadiazine, it can be predicted that lanthanum sulfadiazine will be an efficient, multifunctional and antibacterial thermal stabilizer for PVC. Therefore, the purpose of this study was to explore the possibility of lanthanum sulfadiazine as a novel rare earth thermal stabilizer for PVC. We predicted that LaSD can effectively improve the thermal stability of PVC. In addition, it is very interesting to study the co-stabilization effect of LaSD with other commercial stabilizers on PVC. Therefore, this paper also studied the synergistic stability of LaSD in combination with epoxy soybean oil (ESBO) and calcium stearate (CaSt_2_) on PVC.

## 2. Materials and Methods

### 2.1. Materials

Poly(vinyl chloride) (PVC, SG-5) was purchased from Xinjiang Tianye Group Co. LTD., Shihezi, China. Calcium carbonate (CaCO_3_, 1000 mesh), was purchased from Zhejiang Himpton New Materials Co. LTD., Hangzhou, China. Calcium stearate (CaSt_2_, calcium content: 6.6–7.4%), zinc stearate (ZnSt_2_, zinc content: 10–12%), lead subacetate (LSA, AR), uracil (98%), dioctyl phthalate (DOP, 98%), sulfadiazine (SD, 99%), lanthanum nitrate hexahydrate (La(NO_3_)_3_·6H_2_O, 99%), and epoxy soybean oil (ESBO, AR) were purchased from Aladdin Reagent Company., Shanghai, China. Other chemical reagents used in this study were analytical reagents and the Ca/Zn thermal stabilizer was prepared through CaSt_2_ and ZnSt_2_ with their mass ratio about 1:1.

### 2.2. Synthesis and Characterization of LaSD

Sulfadiazine (3.75 g, 15 mmol), NaOH (0.6 g, 15 mmol) and 100 mL distilled water were mixed into a 250 mL three-necked flask and heated to 75 °C. After stirring and dissolving completely, the 1.0 mol/L lanthanum nitrate solution (5 mL, 5 mmol) was gradually dropped into a three-necked flask through a constant pressure funnel. After stirring the precipitate for 1 h, it was filtered and washed with heated distilled water. The white solids were dried at 140 °C with a yield of 79.8%. The synthetic route of LaSD is shown in Scheme 1.

LaSD was characterized by the crucible thermogravimetric method, element analysis (EA), Fourier transform infrared spectrophotometry (FTIR) and thermogravimetric analysis (TGA). The content of lanthanum in LaSD was determined by the crucible thermogravimetric method, which involved baking it in a muffle oven at 950 °C for 4 h. The carbon, hydrogen, nitrogen and sulfur content in LaSD were ascertained by an elemental analyzer (Euro EA 3000, EA Instruments, Milan, Italy). The molecular structure of the sample was confirmed by Fourier transform infrared spectroscopy (IRPrestige-21, Shimadzu Corp., Kyoto, Japan) in the range of 4000–400 cm^−1^. Thermogravimetric curves were measured through a thermogravimetric analyzer (TG/DTA6300, Seiko Instruments Inc., Chiba, Japan) in an air atmosphere at a heating temperature about 10 °C/min between room temperature and 750 °C.

### 2.3. Preparation of PVC Samples

The preparation of PVC splines included the following steps: firstly, PVC powder resin (50 phr), PVC paste resin (50 phr), dioctyl phthalate (50 phr), calcium carbonate (15 phr) and stabilizer (3 phr) were ground and mixed in a mortar. Secondly, the mixture was poured into a 1.0 mm thick glass mold and plasticized at 140 °C for 40 min. Finally, the cooled PVC sample was cut to the appropriate size (15.0 mm × 20.0 mm × 1.0 mm).

### 2.4. Evaluation of Stabilizing Efficiency

#### 2.4.1. Congo Red Test

According to the standard method for the Congo red test (GB2917-2002), a certain amount of PVC and thermal stabilizer with a mass ratio of about 100:3 was adequately ground in the mortar. Then 3 g of the mixture was weighed and put into a glass test tube [5,6]. Subsequently, the Congo red test paper was inserted into the glass test tube and its lowest edge was 2 cm above the upper surface of the mixture to be tested. Then, the tube was transferred into an oil bath at 180 °C and the stopwatch was started. Finally, the static thermal stability time was confirmed when the color of the Congo red test paper changed from red to blue. Each group of samples was tested three times and the average value was taken as the final result.

#### 2.4.2. Discoloration Test

According to the standard test method for oven discoloration (GB2917-2002), the PVC samples were firstly cut into an appropriate size (15.0 mm × 20.0 mm × 1.0 mm) and then put onto aluminum foil [5,6]. Subsequently, the PVC samples were transferred to a temperature-controlled oven (UF260, Memmert Inc., Schwabach, Germany) at a temperature of 180 °C. The sheets were removed from the temperature-controlled oven every 10 min and the discoloration was recorded using a scanner (LiDE120, Canon Inc., Tokyo, Japan).

#### 2.4.3. Analysis of Thermal Degradation

There were two types of thermal degradation tests:

The first method measured the thermal degradation process of PVC samples with different concentration of thermal stabilizers by using a thermogravimetric analyzer (TG/DTA6300, Seiko Instruments Inc., Chiba, Japan). Each sample was scanned from room temperature to 700 °C under air atmosphere at a heating rate of 10 °C /min.

The second method investigated the thermal decomposition kinetics of PVC samples with different complex thermal stabilizers by also using the thermogravimetric analyzer (TG/DTA6300, Seiko Instruments Inc., Chiba, Japan). Each sample was scanned from room temperature to 400 °C under the air atmosphere at the heating rates of 10, 15, 20 and 25 °C/min. Kinetic parameters such as activation energy (Ea, KJ/mol) were calculated using the Kissinger equation [18]:(1)$$\ln\left(\frac{\beta_{i}}{T_{P}^{2}}\right)=\ln\frac{A_{a}R_{}}{E_{a}}-\frac{E_{a}}{R_{}}\frac{1_{}}{T_{p}}{\left[i=1,2,3,4,5\right]}_{}$$
where R is the gas constant (8.314 mol/K), *β**_i_* is the heating rate, and *T_p_* is the peak temperature on DTA. Equation (1) indicates a linear correlation between ln(*β**_i_*/*T_p_*^2^) and 1/*T_p_*. Finally, *E_a_* can be calculated from the slope (a) on the curves of ln(β_i_/*T_p_*^2^) change versus 1/*T_p_* according to Equation (2).
(2)$$E_{a}=-{aR}_{}$$

#### 2.4.4. Study of the Mechanism of LaSD as PVC Thermal Stabilizer

The mechanism of LaSD as a thermal stabilizer in PVC was studied through the following two experiments [21,22].

The first experiment investigated whether LaSD could be used as a HCl absorbent. Firstly, a certain amount of LaSD was put into a three-neck flask and placed in an oil bath at 180 °C. Secondly, the LaSD was circulated with dry HCl gas for 2 h, and then the treated LaSD was heated in air at 120 °C for another 4 h to remove residual HCl. Thirdly, the treated products were added into deionized water and filtered to obtain the filtrate. Finally, a drop of 0.1 mol/L silver nitrate solution was added into the filtrate. The production of white precipitate in filtrate indicates whether the filtrate contains chloride ions, which further determined whether LaSD could be used as a HCl absorbent.

The second experiment examined whether LaSD could replace unstable chlorine atoms in PVC chains. A certain amount of LaSD and PVC were uniformly mixed in a mortar and plasticized at 180 °C for 10 min in a double-roll mill (LN-160/6, Dongguan Linna Machinery Industrial Co. Ltd., Guangdong, China). The PVC samples were dissolved in THF and filtered to remove the unreacted LaSD. Subsequently, PVC samples were precipitated in methanol and collected by filtration. The purified PVC samples were placed in a temperature-controlled oven for 30 min at 180 °C and characterized by FTIR spectrum.

### 2.5. UV-Vis Spectrum

The PVC samples with a thickness of 1 mm were cut into an appropriate size (30 mm × 30 mm) and placed in a temperature-controlled oven (UF260, Memmert Inc., Schwabach, Germany) at 180 °C. PVC samples were removed from the oven every 40 min and the absorbance of PVC at different aging times was measured by ultraviolet-visible-infrared spectrophotometer (UV-2450, Shimadzu Corp., Kyoto, Japan) [25].

## 3. Results

### 3.1. Characterization of LaSD

Lanthanum sulfadiazine (LaSD) was synthesized in our laboratory under alkaline conditions. The content of lanthanum in LaSD obtained by the crucible thermogravimetric method was 15.21%. The elemental analysis results are shown in Table 1; it was confirmed that the mass ratio of La:C is 15.21:40.01 and it was inferred that its atomic concentration ratio of La:C is about 1:30. Based on the experimental results, it was inferred that the molecular formula of the product is La(C_10_H_9_N_4_O_2_S)_3_ [26,27]. LaSD is used to represent La(C_10_H_9_N_4_O_2_S)_3_ in this paper.

The thermal degradation process of LaSD at room temperature to 750 °C was characterized by thermogravimetric analysis (TGA). Figure 1 shows the TGA curve of LaSD and it can be seen that there is a tiny mass loss within the temperature range of 25–250 °C, which is mainly attributed to the adsorbed water [28]. It can also be seen from the figure that LaSD is relatively stable and its mass loss does not exceeded 1.0 wt % from room temperature to 200 °C. This indicates that LaSD is relatively stable in the processing temperature range of 160–200 °C for PVC and can be used as a stabilizer for PVC [28]. In addition, according to Figure 1, the TGA curve of LaSD shows that the decomposition residue is 16.67% (calcd. 18.39%) which also proves that the molecular formula of La(C_10_H_9_N_4_O_2_S)_3_ is correct.

The infrared characteristic peaks of 4000–400 cm^−1^ were characterized by Fourier transform infrared spectroscopy (FTIR) and the FTIR spectrum of LaSD and SD are shown in Figure 2. As can be seen from Figure 2, the peak value of 3426.5, 3353.8 and 1582.9 cm^−1^ represent the stretching vibration peak and bending vibration peak, respectively, of NH_2_ [29]. These characteristic peaks still exist after the formation of LaSD, indicating that there is no reaction between La^3+^ and NH_2_. Moreover, an infrared characteristic peak is added at 418.4 cm^−1^ after the formation of lanthanum salt, which is attributed to the La-N characteristic peaks [30]. So, it can be proved that LaSD has been successfully synthesized.

### 3.2. Effects of Different Thermal Stabilizers on Stabilizing PVC

The Congo red method was used to evaluate the influence of different thermal stabilizers on stabilizing PVC, and the results are shown in Figure 3 [6,26]. As can be seen from the figure, the thermal stability time of pure PVC is about 390 s and the thermal stability was improved to varying degrees when the traditional commercial thermal stabilizers LSA, uracil, Ca/Zn and ZnSt_2_ are added into PVC, respectively. Among them, LSA has the most significant impact on the thermal stability time of PVC, which can be reached in 1120 s. After the addition of LaSD, the thermal stability time of PVC can be reached in 1330 s, which may be because LaSD has a strong ability to absorb HCl [6,26]. Thus, it can effectively inhibit the thermal degradation process of PVC and significantly improve its long-term stability. Therefore, LaSD can be used as an effective thermal stabilizer for PVC.

The oven discoloration method was also used to evaluate the influence of different thermal stabilizers on stabilizing PVC, and the results are shown in Figure 4 [31]. As can be seen from Figure 4, the PVC sample with ZnSt_2_ has a good initial color, but it turned completely black after 10 min. The main reason is that ZnSt_2_ can inhibit discoloration by replacing unstable chlorine atoms in the PVC chain. However, ZnCl_2_ will be produced in this process and catalyzes PVC degradation. PVC samples suddenly turn black with the increase in the concentration of ZnCl_2_, which is called the “zinc burning” phenomenon [31]. For PVC samples with Ca/Zn(1:1), the complete discoloration time is prolonged to 60 min, which is caused by the reaction of ZnCl_2_ and CaSt_2_ to produce CaCl_2_ [32]. It can also be seen from the figure that the thermal stability of PVC is greatly improved after the addition of LSA and uracil, which may be because both of LSA and uracil can absorb HCl, thus inhibiting the degradation of PVC and improving its long-term stability [21,26]. Compared with the above thermal stabilizers, LaSD not only inhibits the initial staining, but also effectively improves its long-term stability. This may be because LaSD not only replaces unstable chlorine atoms but it can also absorb HCl, thus, it can improve the thermal stability of PVC.

In order to explore the complete thermal degradation process of LaSD in PVC, different concentrations of PVC/LaSD samples were evaluated by thermogravimetric analysis, and the results are shown in Figure 5 [32,33]. As can be seen from Figure 5, the thermal degradation process of PVC can be divided into two stages of thermal weightlessness. The first thermal weightlessness stage is mainly attributed to the removal of hydrogen chloride from the PVC main chain, thus forming a conjugated polyene structure. The second thermogravimetric stage is mainly attributed to the fracture of conjugated polyene structures to form small molecular weight linear or cyclic hydrocarbons, as shown in Scheme 2 [32,33].

Table 2 summarizes the initial decomposition temperature (T_5%_ and T_10%_), the fastest decomposition temperature (T_f_) and the first stage of thermal weight loss (W_f_) of PVC/LaSD samples with diverse concentrations during the thermal degradation process. As can be seen from Table 1, PVC/LaSD samples with diverse concentrations are markedly improved in T_5%_, T_10__%_ and T_f_ compared with pure PVC, and these values are increased with the increase in concentration. This indicates that LaSD can positively promote the thermal stability of PVC samples [34]. On the other hand, the W_f_ value of PVC/LaSD samples gradually decreases with the continuous increase in LaSD concentration, which indicates that LaSD can absorb HCl released by PVC degradation [34]. These results are consistent with the results of the Congo red test and discoloration test.

### 3.3. Study of the Thermal Stability Mechanism of LaSD in PVC

In order to propose a possible mechanism to explain the good thermal stability efficiency of LaSD on PVC, we studied the action mode of LaSD as PVC thermal stabilizer through the two experiments in 2.4.4 [34]. In the first experiment, white precipitate appears in the transparent filtrate when 0.1 mol/L of silver nitrate solution is added. This indicates that the filtrate contains chloride ions, which can further verify that LaSD can react with HCl at 180 °C [29]. Furthermore, Figure 6 shows the infrared spectra of LaSD before (a) and after (b) HCl treatment at 180 °C. As can be seen from the figure, a C-Cl characteristic peak appeared in 782.3 cm^−1^ after HCl treatment [35], and the peak value of 3426.5, 3353.8 and 1582.9 cm^−1^ which represent the stretching vibration peak and bending vibration peak, respectively, of NH_2_ all disappeared after HCl treatment. These results suggest that LaSD can act as an HCl absorber [29]. Figure 7 shows the infrared spectrum of pure PVC, LaSD and purified PVC. As can be seen from Figure 7, the infrared spectrum of purified PVC, which contained the characteristic peaks appears in LaSD [5,26]. Therefore, we believe that LaSD can replace unstable chlorine atoms on the PVC chain.

According to the above analysis results, a conceivable mechanism of LaSD to stabilize PVC is proposed and described in Scheme 3. It was found that unstable chlorine atoms are separated from the PVC chain under the attraction of La^3+^ ions to form LaCl_3_ [14,15]. Subsequently, the sulfadiazine negative ion intermediate is attached to the carbocation on the PVC chain (**1**–**6**). This suggests that LaSD can ulteriorly postpone dehydrogenation of the zipper by replacing the unstable chlorine on the PVC chain. In addition, the sulfadiazine negative ion intermediates on PVC chains have a strong capacity for adsorbing HCl, which are released by degradation of PVC (**7**–**8**). Therefore, LaSD displays higher initial whiteness and longer-term stability for stabilizing PVC than other stabilizers.

### 3.4. Influence of LaSD/ESBO on Stabilizing PVC

Figure 8 and Figure 9 show the Congo red test and oven discoloration test results of PVC samples with different proportions of LaSD/ESBO complex stabilizer. The results of the Congo red test (Figure 8) show that the thermal stability time of PVC/LaSD and PVC/ESBO samples are 1330 s and 936 s, respectively. This indicates that both LaSD and ESBO have good thermal stability for PVC. In addition, the LaSD/ESBO complex thermal stabilizers with different proportions reveal a higher thermal stability time on stabilizing PVC than that when they are used alone. Moreover, the thermal stability time of PVC samples are increased first and then decreased with increasing the proportion of LaSD/ESBO. The thermal stability time reaches its highest when the ratio of LaSD/ESBO is about 1.2/1.8, which indicates that there are good synergies between LaSD and ESBO [26,36]. Furthermore, the oven discoloration test (Figure 9) shows that the results are basically consistent with the Congo red test. The sample has the highest initial whiteness and long-term stability when the LaSD/ESBO ratio is about 1.2/1.8, which further proves that LaSD and ESBO have a good synergistic effect. This can be chiefly due to LaSD absorbs HCl through NH_2_ while ESBO absorbs HCl through the epoxy group, so they do not compete with each other in absorbing HCl [26,36]. Therefore, ESBO can be used as an auxiliary stabilizer for LaSD.

### 3.5. Influence of LaSD/ESBO/CaSt_2_ on Stabilizing PVC

It is of great significance to study the synergistic effect between the main stabilizer and the auxiliary stabilizer [26,36]. The above results show that LaSD and ESBO have a good synergistic effect, which can significantly restrain the degradation of PVC and enhance its thermal stability. CaSt_2_ is a long-term auxiliary thermal stabilizer and it cannot effectively restrain PVC discoloration when applied alone, so, it needs to be used together with other stabilizers [26,36]. Here, in order to further improve the long-term stability of PVC, the CaSt_2_ is introduced into the LaSD/ESBO and when combined it turns into a ternary complex stabilizer. The Congo red test and oven discoloration test results of PVC/LaSD/ESBO/CaSt_2_ samples with different mass ratios are shown in Figure 10 and Figure 11. According to the comparison of the Congo red test results in Figure 8 and Figure 10, it can be seen that the PVC/LaSD/CaSt_2_ sample shows a higher thermal stability time (about 2006 s) than that of the PVC/LaSD/ESBO sample (about 1412 s) using the same proportions, which indicates that LaSD/CaSt_2_ has a better synergistic effect on stabilizing PVC compared with LaSD/ESBO. As can be seen from the Congo red test in Figure 10, the overall thermal stability time of PVC/LaSD/ESBO/ CaSt_2_ samples is higher than that of PVC/LaSD/ESBO. This result shows that the addition of CaSt_2_ also has a significant effect on improving the thermal stability time of the PVC system. However, the thermal stability time of PVC/LaSD/ESBO/CaSt_2_ samples are first increased and then decreased with increasing the proportion of LaSD/ESBO and the thermal stability time reaches its highest when the proportion of LaSD/ESBO/CaSt_2_ is about 1.8:0.6:0.6. This may be because the synergistic effects between LaSD and CaSt_2_ are dominated by the main role of the ternary composite stabilizer when the content of LaSD is relatively high, so, it is conducive to the increase in thermal stability time [26,36]. However, the antagonistic effect between CaSt_2_ and ESBO plays a major role when the content of ESBO increases while the content of LaSD decreases. They compete to absorb HCl in the process of PVC degradation and this results in decreasing the thermal stability of PVC [36]. The results of the Congo red test are further verified by the oven discoloration test (Figure 11).

### 3.6. Thermal Decomposition Kinetics

In order to compare the thermal stability effects of LaSD, LaSD/ESBO, LaSD/ESBO/CaSt_2_ on PVC, the three samples which had the best thermal stability effects from the above experiments were selected for TGA testing at diverse heating rates [37,38]. The test conditions involved heated from room temperature to 400 °C in the air atmosphere with heating rates of 10, 15, 20 and 25 K/min. Figure 11 shows the TGA and DTG curves of PVC/LaSD, PVC/LaSD/ESBO and PVC/LaSD/ESBO/CaSt_2_. From the curve, it could be seen that the thermogravimetry stage of PVC from room temperature to 400 °C is principally to remove the HCl from the main PVC chain, and subsequently, to form conjugated polyene structures [37,38]. In order to quantitatively assess the influence of LaSD, LaSD/ESBO and LaSD/ESBO/CaSt_2_ as stabilizers on stabilizing PVC, the Kissinger method was used to calculate and compare the first stage activation energy (E_a_) of the PVC/LaSD, PVC/LaSD/ESBO and PVC/LaSD/ESBO/CaSt_2_ samples [37,38]. According to the Kissinger method, the relationship between the maximum reaction rate and temperature is described by Equation (1) and the kinetic parameters of the three thermal stabilizers are shown in Figure 12, Figure 13 and Table 3. It can be seen that the order of the Ea value is PVC/LaSD/ESBO/ CaSt_2_ (119.9 kJ/mol) > PVC/LaSD/ESBO (112.1 kJ/mol) > PVC/LaSD (106.6 kJ/mol). Therefore, according to the kinetic analysis, LaSD/ESBO/CaSt_2_ can be applied as a kind of highly efficient long-term thermal stabilizer for PVC. The results of thermal decomposition kinetics are consistent with those of the Congo red test and oven discoloration test.

### 3.7. UV-Vis Spectrum Test

It is well known that the aging and discoloration of PVC is primarily caused by the reaction of dehydrogenation of the zipper to eliminate HCl and the formation of polyene structures on the main PVC chain [14]. Previous research had shown that a certain relationship exists between the concentration of conjugated double bonds and the UV-vis absorption wavelength and the maximum absorption wavelength in the UV-Vis spectrum is highly dependent on the unsaturated conjugated sequence (n). In addition, the UV-vis absorption wavelength is gradually increased in the visible region when the aging and discoloration of PVC are aggravated [14,29]. Therefore, it can be considered that the greater the absorption wavelength in the visible region represents the higher the concentration of the conjugate double bond in the PVC chain. So, the degree of PVC aging can be deduced through this method. In order to research the concentration variation of conjugate double bonds of PVC/LaSD, PVC/LaSD/ESBO and PVC/LaSD/ESBO/CaSt_2_ samples in the aging process, these samples were placed in the temperature-controlled oven with a temperature of 180 °C. Subsequently, the samples were taken out from the oven at 0 min, 40 min, 80 min and 120 min and measured by ultraviolet-visible-infrared spectrophotometer. Figure 14 exhibits the results of the UV-vis absorption curves and it can be seen that PVC/LaSD, PVC/LaSD/ESBO and PVC/LaSD/ESBO/CaSt_2_ samples display analogical curves in the visible light absorption region at 0 min. However, these samples show varying degrees of change in the visible light region after roasting for 40 min, and the order of change is PVC/LaSD/ESBO/CaSt_2_ > PVC/LaSD/ESBO > PVC/LaSD. The change is even more prominent after 80 min and 120 min. Hence, from the above results, it can be expressly inferred that the sequence of the thermal stability effects of the three stabilizers on PVC is LaSD/ESBO/CaSt_2_ > LaSD/ESBO > LaSD [14,29]. These results are consistent with the above thermogravimetric kinetic analysis results. The UV-vis absorption experiment also substantiates that the synergistic effects among LaSD, ESBO and CaSt_2_ are distinct and they can improve the thermal stability of PVC. Therefore, the results again show that LaSD/ESBO/CaSt_2_ can be applied as a kind of highly efficient long-term thermal stabilizer for PVC.

## 4. Conclusions

A new kind of rare earth thermal stabilizer (LaSD) was prepared by the double decomposition reaction of sulfadiazine, lanthanum nitrate and sodium hydroxide. Its molecular structure was ascertained to be La(C_10_H_9_N_4_O_2_S)_3_ by EA, FTIR and TGA. The Congo red test, oven discoloration test and TGA test indicate that LaSD can prolong the thermal stability time and long-term stability of PVC through absorbing the HCl released by PVC and replacing the unstable chlorine atoms in the PVC chain during the degradation process. Furthermore, LaSD/ESBO/CaSt_2_ complex thermal stabilizer displays outstanding synergistic effects on stabilizing PVC and the thermal stability achieves optimum when the proportion of LaSD/ESBO/CaSt_2_ is 1.8:0.6:0.6. Therefore, as a newly-fashioned non-toxic stabilizer, the LaSD/ESBO/CaSt_2_ complex stabilizer is expected to supersede the traditional thermal stabilizer used during PVC processing in the future.

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
