# Peer review of "Development and Investigation of Lanthanum Sulfadiazine with Calcium Stearate and Epoxidised Soyabean Oil as Complex Thermal Stabilizers for Stabilizing Poly(vinyl chloride)"

_polymers, 2019, doi:10.3390/polym11030531_

Round 1

Reviewer 1 Report

This work regarding "Development and Investigation of Lanthanum Sulfadiazine with Calcium Stearate and Epoxidised Soyabean Oil as Complex Thermal Stabilizers for

Stabilizing Poly(vinyl chloride)" discusses Lanthanum sulfadiazine (LaSD) used as a thermal stabilizer to stabilize PVC. I give my useful comments and feedback to the author to improve the quality of the manuscript after careful reviewing: 

 1. Authors should state what new scientific contribution is contained
 in their manuscript compared to any previous articles published by other
 researches. In other words, the novelty of the research must be pronounced
 clearly and explicitly in the “Abstract” or in the last paragraph of "Introduction".

2. Figure 3 does not seem to be applicable please indicate from where
the data is counted
3. FTIR spectrum is poorly legible
4. Schemes are poorly described
5. Figure 12 should be reduced, too much unnecessary data
6. I do not understand the aging time of PVC, 40, 80, 120  minutes.
PVC is very resistant to aging and time seems short (Figure 14)
7. 2. Formatting of the manuscript need to be improved such as citation,
table and figure.
This paper may be considered to publication in Polymers,
  but only after major revision.

Author Response

Dear Editors and Reviewers:

Thank you very much for your letter and for the reviewers’ comments concerning our manuscript entitled “Development and Investigation of Lanthanum Sulfadiazine with Calcium Stearate and Epoxidised Soyabean Oil as Complex Thermal Stabilizers for Stabilizing Poly(vinyl chloride)” (ID: polymers-466163). From the comments, I am clearly observed that my dear editors and reviewers are all very responsible and respected. Those comments are all valuable and very helpful for revising and improving our paper, as well as the important guiding significance to our researches. We have studied comments carefully and have made correction which we hope meet with approval. Revised portion are marked in red in the paper. The main corrections in the paper and the responds to the reviewer’s comments are in the attachment

Reviewer 2 Report

In my opinion the results obtained by the authors are valuable and are worth to be published in Polymers. The authors proved that a new non-toxic stabilizer (LaSD/ESBO/CaSt2) may replace in the future the traditional thermal  stabilizer during PVC processing.

In my opinion, the article has some shortcomings and needs to be improved before being accepted for publication.

Dear authors, please refer to the following suggestions and comments:

Major comments

Comment 1.

In this year in Polymers you published the article titled „Synthesis and Study of Zinc Orotate and its Synergistic Effect with Commercial Stabilizers for Stabilizing Poly(Vinyl Chloride)”. In this article you described the application of CaSt2/ZnOr2/PER (calcium stearate/ zinc stearate/ pentaerythritol) for the stabilizing of PVC. In the last sentence of this work we can find the statement „…CaSt2/ZnOr2/PER can be applied as a potential high efficiency complex thermal stabilizer for PVC in the future”. In the case of reviewed paper we can find the similar conclusion „…LaSD/ESBO/CaSt2 complex stabilizer is expected to replace the traditional thermal stabilizer during PVC processing in the future.” In both cases you used the similar parameters to describe thermal stability of modified PVC. In my opinion you have to compare the influence of both stabilizers (CaSt2/ZnOr2/PER and LaSD/ESBO/CaSt2) on the parameters of new materials. Please add such comparison to the article.

Comment 2.

As you wrote in the introduction poly(vinyl chloride) is widely use due to its excellent physical and chemical properties, including fire resistance, wear resistance, best comprehensive mechanical properties, transparency, electrical insulation, chemical resistance, heat resistance, sound insulation and other advantages. Your investigation was focused on evaluation of thermal stability of PVC stabilized by means of lanthanum sulfadiazine with calcium stearate, and epoxidized soybean oil. Are you able to give information about the influence of these stabilizers on other physical and chemical properties of stabilized in this way PVC. These parameters determine the possibility of using this material in various branches of industry.

Comment 3.

Section 3.2.The information about preparation of PVC stabilized by means of LSA, uracil, Ca/Zn, ZnSt2 should be added. The method of PVC modification (e.g. stabilizer content) could have a significant influence on the results.

Minor comments:

Line 19 All abbreviations should be expanded, even these well known. Please expand the abbreviation PVC

Line 116 It is not necessary to give the information about manufacturer of muffle oven

Line 116 The name of the oven manufacturer can be removed

Line 174 Add space between “mill and (LN-160/6 ...)”

Line 175 Remove space between “...China) and .”

Line 204 Add “differential thermogravimetry” after DTG

Line 220 Please expand the abbreviation LSA

Author Response

(The authors gave the same response as above.)

Round 2

Reviewer 1 Report

 I Accept in present form